# Increasing CO$_2$ flux at Pisciarelli, Campi Flegrei, Italy

Manuel Queißer[1], Domenico Granieri[2], Mike Burton[1], Fabio Arzilli[1], Rosario Avino[3], Antonio Carandente[3]

[1]School of Earth, Atmospheric and Environmental Sciences, University of Manchester, Oxford Road, Manchester M13 9PL, UK

[2]Istituto Nazionale di Geofisica e Vulcanologia (INGV), Sezione di Pisa, 50126 Pisa, Italy

[3]Istituto Nazionale di Geofisica e Vulcanologia, Osservatorio Vesuviano, Napoli, 80124, Italy

*Correspondence to*: Manuel Queißer (manuel.queisser@manchester.ac.uk), Tel.: +44(0)161 2750778, Fax.: +44(0)161 306 9361

**Abstract.** Campi Flegrei caldera is located in the metropolitan nucleus of Naples (Italy), and has been undergoing different stages of unrest since 1950, evidenced by episodes of significant ground uplift followed by minor subsidence, increasing and fluctuating emission strengths of water vapor and CO$_2$ from fumaroles, and periodic seismic crises. We deployed a scanning laser remote sensing spectrometer (LARSS) that measures path integrated CO$_2$ concentrations at the Pisciarelli area in May 2017. The resulting mean CO$_2$ flux is 578 ± 246 t d$^{-1}$. Our data suggest a significant increase in CO$_2$ flux at this site since 2015. Together with recent geophysical observations, this suggests a greater contribution of the magmatic source to the degassing and/or an increase of permeability at shallow levels. Thanks to the integrated path soundings, LARSS may help to give representative measurements from large regions containing different CO$_2$ sources, including fumaroles, low-T vents, and degassing soils, helping to constrain the contribution of deep gases and their migration mechanisms towards the surface.

## 1 Introduction

Of all the volcanic calderas in the world the ~12 km wide Campi Flegrei (CF) in southern Italy is arguably the one with the highest destructive potential, since it is in a state of unrest and located within an urban area of over 2 million residents, with Naples being the largest urban nucleus in the area (Fig. 1a). Its last eruption dates back to 1538 (Dvorak and Gasparini, 1991). Ever since, CF underwent various series of new, rather swift uplifts (bradyseisms), indicating unrest followed by a decrease in ground level usually at a much slower rate (Chiodini et al., 2010; Troiano et al., 2011; D'Auria, 2015; De Natale et al., 2017). Since the last energetic unrest of 1982-84 the caldera is subject to intense geophysical and geochemical monitoring, with greatest interest for the Solfatara crater, in the center of CF, and for the Pisciarelli area, on the eastern outer slope of Solfatara. Around 2005 a new net uplift, although at a relatively slow rate, has commenced. At Pisciarelli, where the more recent low-energetic seismic swarms are localized (D'Auria et al., 2011), the fumarole temperature increased from below 100°C in 2005 to ~ 115°C in 2015. The amount of water vapor has increased visibly and the strongly degassing area has been considerably enlarged in the past few years (Chiodini et al., 2015). Given these major signs as well as other signs,

mainly related to fluid geochemical variations at the fumaroles of Solfatara (Chiodini et al., 2015, 2016), national civil protection authorities have changed the state of CF from green (quite) to yellow (scientific attention).

As all calderas, CF represents a complicated makeup that includes a magmatic plumbing system up to a depth of ~ 8 km (Bodnar et al., 2007; Zollo et al., 2008; Vitale et al. 2014; Moretti et al., 2017), feeding the overlying hydrothermal system (Chiodini et al., 2010; Troiano et al., 2011; De Siena et al., 2017a) through an intricate network of fractures (Zollo et al., 2008; De Siena et al., 2010; Byrdina et al., 2014). A clear picture of the feeding mechanisms and its dynamics is one of the central open questions of CF and subject to ongoing debate. There is a broad consensus among researchers that injections of deep, hot, and oxidized fluids into the hydrothermal system of CF causes increased $CO_2$ soil degassing (Cardellini et al., 2016), increased $CO_2$ content in the fumarole discharges (with consequent decreasing trends of $H_2O/CO_2$ or $H_2S/CO_2$ ratios) and ground uplift (Caliro et al., 2007; Chiodini et al., 2012; Aiuppa et al., 2013). As a matter of fact, there is a fair correlation between soil/fumarole $CO_2$ degassing strength and episodes of ground uplift (D'Auria, 2011; Chiodini et al., 2012, 2016) following this order: uplift, and months later an increase in $CO_2$ relative to other gases. There appear to be two logical main causes for this:

   i)     An increase in supply of fluids and associated thermal energy into the hydrothermal system for depressurization of the magmatic source (Allard et al., 1991; Chiodini et al., 2016). This increased supply is thought to stem from either the ~ 8 km deep main magma reservoir (Bodnar et al., 2007; Zollo et al., 2008; Moretti et al., 2017), or from a contribution of a magma batch that intruded the shallow subsurface (~ 3-4 km depth) concomitantly with the 1982–1984 unrest episode (Chiodini et al., 2010; Caliro et al., 2014) and periodically rejuvenated by arrivals of deep more primitive magma (Bagagli et al., 2017) and/or

   ii)    an increase in permeability at shallow levels, i.e., above the hydrothermal reservoir (Todesco et al., 2003; Acocella et al., 2015; Piochi et al., 2015).

Discriminating within those mechanisms is out of the scope of this study, but any insights towards a better understanding of these processes are important to improve early warning and civil protection measures at the CF area. Measuring emission rates (fluxes) of $CO_2$ provides an additional way to assess the hazard at CF. The fumarole area of Piscarelli, approximately located in the center of the CF caldera (Fig. 1a) and recently scene of drastic changes in its activity, is a prime geochemical sampling spot to learn about the volcanic processes taking place beneath CF. A spatially integrated measurement of $CO_2$ flux that accounts for all possible $CO_2$ vents and diffuse degassing is desirable to obtain a quantitative picture of $CO_2$ degassing, but has only been done a few times after 2012 at Pisciarelli (Pedone et al., 2014; Aiuppa et al., 2015; Queißer et al., 2016a). To increase the number of observations it was decided to revisit CF 14 months after the last such measurement (Queißer et al., 2016a) and re-measure $CO_2$ fluxes.

## 2 Materials and Methods

The $CO_2$ concentrations needed to estimate the $CO_2$ flux are commonly sampled at points, which may miss out sources, such as smaller fumarolic discharges (Chiodini et al., 2015). On the other hand, point measurements are very precise and valuable in characterizing local degassing elements, such as fractures. Path integrating, scanning gas measurement techniques, on the other hand, may add value by providing a spatially comprehensive measurement. To attempt a spatially inclusive measurement of all possible sources of $CO_2$, diffusive soil and vented degassing, we used a laser remote sensing spectrometer (LARSS), developed in the ERC proof-of-concept project CarbSens. Combined with point measurement techniques, such as accumulation chambers, LARRS may help to yield a more complete picture of degassing. It represents a further miniaturization of a similar system developed in the ERC project $CO_2$Volc. The instrument and its working principle are detailed elsewhere (Queißer et al., 2017). Only a brief overview is given therefore. LARSS consists of a main unit and a transmitter/receiver unit (TX/RX unit, Fig. 1b). The latter comprises of the telescope, transmitter and an integrating sphere for power reference measurement. It is portable (mass: 10 kg main unit + 6 kg TX/RX unit), which allows it to be transported easily and set up at any kind of surface, such as house roofs or airplanes.

The $CO_2$ absorption line at 1572.335 nm ($R$16 transition) is sampled at 40 wavelengths by sweeping the emission wavelength of a diode laser. The laser light is amplified, transmitted, backscattered at a topographic target and received by the telescope. After the detected signal is digitized, the optical transmittance of the telescope's viewing path is deduced for each of the 40 wavelengths. A model absorption spectrum is fitted to the 40 measured transmittances, resulting in a best estimate of the path averaged $CO_2$ column density (in $m^{-2}$). The path length may be up to 2 km. Profiles of $CO_2$ concentrations, i.e., $CO_2$ concentrations versus angle, are attained by scanning the TX/RX unit across a degassing plume (see Queißer et al. (2016a) for details on scanning geometry). Along with the plume transport speed these profiles are then used to obtain $CO_2$ fluxes, following

$$\Phi_{CO_2} = u \frac{M_{CO_2}}{N_A} \Delta\beta \sum_{plume} r_i N^{col}_{pl}(r_i),$$

(1)

where $u$ refers to the component of the plume transport speed perpendicular the plane of the $CO_2$ concentration profile, i.e., the component perpendicular to the plane of the scan. $M_{CO_2}$ is the molar mass of $CO_2$ (in kg $mol^{-1}$) and $N_A$ is Avogadro's constant (in $mol^{-1}$). $\Delta\beta$ is the constant scan angle increment. $N^{col}_{pl}$, the background corrected, or in-plume column density of $CO_2$, is retrieved by subtracting the total $CO_2$ column density by the ambient $CO_2$ column density measured outside the plume, i.e., $N^{col}_{pl}(r_i) = N^{col}(r_i) - N^{col}_{bg}(r_i)$ , where $N^{col}(r_i)$ is the total column density as measured, and $N^{col}_{bg}(r_i)$ depicts the ambient column density. The ranges $r_i$ are measured with a range finder LIDAR aligned with the telescope. For

convenience and display purposes, if meteorological data are available, column densities may be converted to path averaged mixing ratios (in ppm) as detailed in Queißer et al. (2017). The plume speed is retrieved by digital video tracking of the plume of condensed water vapor as described in Queißer et al. (2016a).

**3 Results**

The data presented here are a subset of data acquired during a campaign probing $CO_2$ at the Pisciarelli-Solfatara area between 24[th] and 26[th] of May 2017. LARSS was placed on the roof of the *Tennis Hotel*, located ~ 320 m east of the Pisciarelli fumaroles, offering an unobstructed view on the complete fumarole degassing activity. Between 17:07 and 18:04 local time on the 24 May 2017, 9 lateral angular scans were performed, out of which 6 are displayed in Fig. 2. A step motor
rotated the TX/RX unit between 257.4° and 243.4° (Fig. 1b) with a velocity of 2.5 mrad s$^{-1}$, corresponding to a lateral section of ~80 cm at the fumarole area per data point. $\Delta\beta$ was retrieved by multiplying the scanning angular speed with the time between subsequent measurements as recorded in the time stamps of the raw data files. Each scan took around 90 s. Meteorological data (temperature, pressure, humidity) were recorded using a Kestrel portable meteorological station placed next to LARSS. $N^{col}_{bg}$ were measured by a scan upwind, outside of any gas plume, using a hill range between 700 and 900
m distance as target. The corresponding column averaged $CO_2$ mixing ratio was found to be 499 ppm. For comparison, two in-situ measurements with a LI-COR analyzer were performed at points near the optical paths of LARSS, yielding $CO_2$ mixing ratios of 550 ppm and 560 ppm, respectively. These are remarkably high $CO_2$ concentrations, given that the wind came from the sea (South). The proximity of the measurement points to the road and the dense network of roads in that area may well cause these values (Schmidt et al., 2014). Consequently, $N^{col}_{bg}$ corresponding to an ambient $CO_2$ mixing ratio of
499 pm ± 61 ppm were considered.

Highest $CO_2$ concentrations were usually detected near the center of the probed area (near 250°, Figs. 2b to g). This main plume reveals a fine structure, suggesting three sub peaks, which could be related to three main vents in very close proximity to each other identified by Pedone et al. (2014). The highest column averaged $CO_2$ mixing ratio measured was 1777 ppm (Fig. 2e), which is, however, associated with a relatively large uncertainty of 236 ppm (1 STD). Note that Pedone
et al. (2014) measured a peak value of 1444 ppm in early 2013 at approximately the same location. Elevated concentrations also occurred towards the southern edge of the probed area (~ 21 m south of the main plume), at the slope. The corresponding peak repeatedly arose near 246° (especially Figs. 2b, c, d and g).

Uncertainties of path averaged $CO_2$ mixing ratios were usually between 2% and 5% or 10 to 30 ppm (associated with a path averaged detection limit of ~10000 ppm.m). The main source of uncertainty was the contribution of the
instrument itself (baseline drift) and the fitting error. The latter had been significantly improved (roughly halved) by

increasing the number of sampled wavelengths from 20 to 40 recently. A detailed description of influences of various error sources is provided in Queißer et al. (2017). In-plume $CO_2$ concentrations found were mostly between 500 and 4000 ppm, with peaks around 6000 ppm, and agree well with those measured by the fixed in-situ station (Figs. 2b to g). In-plume concentrations had associated uncertainties naturally larger than those of the column averaged values, that is, typically between 4% and 15%, or around 150 ppm. Local wind eddies may lead to local maxima of $CO_2$ concentrations and may also explain the shift in the global concentration maximum after Fig. 2d, suggesting a generally "wobbly" character of the $CO_2$ plume.

The measured vertical plume speed component was 0.65 m s$^{-1}$ (min 0.28 m s$^{-1}$, max 1.05 m s$^{-1}$) until 17:43:47 and 0.80 m s$^{-1}$ (min 0.31 m s$^{-1}$, max 1.37 m s$^{-1}$) after that. The plume speed uncertainties were calculated from the student *t*-variance as detailed in Queißer et al. (2016a). Given the complex terrain and the fact that the measurement was performed close to the ground the velocity field across the scanned plume was generally not constant, in addition to temperature variations causing different plume speeds across the plume. The corresponding variability has been accounted for by tracking different paths of propagating water vapor across the plume and using the variability in the error estimation. Plume speed is in fact one of the main sources of uncertainty, adding an uncertainty of the order of 30% to the flux.

Table 1 shows the flux values computed using Eq. (1), with a mean value of 6.7 ± 2.9 kg s$^{-1}$ (578 ± 246 t d$^{-1}$). As noted in previous measurements at Pisciarelli, the measured fluxes fluctuate by over 100% over the course of minutes (Aiuppa et al., 2015; Queißer et al., 2016a). However, an observational window of 1h length reflected the same variability as an 8h long window (Aiuppa et al., 2015). The rigorous error assessment, i.e., taking all relevant error sources into account, including conservative systematic errors estimates, led to a rather high uncertainty of the flux values. The conservatively chosen uncertainty of the ambient $CO_2$ concentration, an order of magnitude higher than usual, accounts for between 20% and 70% (depending on the profile) of the flux uncertainties presented in Table 1. The other chief source of flux uncertainty is the plume speed, which, depending on the scan, caused an increase in error by the same magnitude.

The mean flux of 6.7 kg s$^{-1}$ corresponds to the complete extension of the scan, that is, the vegetation free zone of ~ 70 m in lateral diameter (Fig. 1c). When integrating over the central area only (between 252.5° and 247.0°), roughly including the aforementioned 3 major vents, the mean flux obtained is 284 ± 107 t d$^{-1}$ and is compatible with the estimated area-integrated value from the in-situ automated flux measurement station FLXOV3 (Fig. 3). This may explain the offset between the fluxes of this work and FLXOV3. Focusing on the main vent area only, however, neglects persistent degassing features, such as at the southern edge of the fumarole area, as well as diffusive soil degassing taking place within the scanned sector (Caliro et al., 2007). This spatially comprehensive character of the measurement is one of the main merits of the remote sensing technique applied here.

## 4 Discussion

The soil $CO_2$ flux at CF is known to have increased in magnitude and spatial extension since 2005 (Cardellini et al., 2016) as well as the $CO_2$ content in CF high-T fumaroles (Chiodini et al., 2010; 2016). Figure 3 suggests a slight acceleration in $CO_2$ degassing from the soils of Pisciarelli since about 2009 (FLXOV3 series) confirmed by post-2012 $CO_2$ measurements integrated over the whole exhaling area, which fairly coincides with the observed acceleration in ground uplift. The similarity between the uplift and degassing trends suggests that both processes are intrinsically related. In fact, the preferential exsolution of $CO_2$ from the deep magmatic body due to its low solubility at high pressure implies an associated release of $H_2O$ simultaneously to $CO_2$ output (Chiodini et al., 2001) or when $CO_2$ is completely exhausted in the magma (Chiodini et al., 2016). In any case, the participation of $H_2O$ in the degassing process results in a very efficient mechanism to convey heat from depth to the hydrothermal system and the overlying rocks, favoring thermally-induced dilation (ground deformation) and enhancing the permeability of fluids flowing through them (greater degassing at the surface). Recent findings indeed point towards an impulsive influx of hot magmatic fluids into the hydrothermal system as a possible source mechanism at CF that eventually cause the observed geophysical and geochemical time series, including the present one (Chiodini et al., 2017).

The inspection of Fig. 3 confirms this general scheme although the $CO_2$ fluxes measured at Pisciarelli in May 2017 (this study) and in March 2016 (Queißer et al., 2016a) seem to suggest an increase at a larger rate than the observed uplift would imply. In particular, the latest available data, up to April 2017, suggests a deceleration of ground uplift at CF as of 2016 (Fig. 3 and in more detail INGV, 2017), which, as far as the resolution of our data permits to say, is not accompanied by a leveling out of degassing strength.

Our results related to the $CO_2$ degassing are compatible with findings, which state that the elastic rock matrix of CF is transitioning to inelastic behavior under long-term stress accumulation, accompanied by a permeability increase of the shallow crust, disguising any direct indicator of unrest, such as rapid ground uplift or enhanced seismicity (Bodnar et al., 2007; Di Luccio et al., 2015; Kilburn et al., 2017). In line with this prospect is a clear seismic velocity decrease since 2012 (Zaccarelli and Bianco, 2017), which could be due to, for instance, a softening bulk or increase in $CO_2$ saturation in the CF aquifer (Queißer and Singh, 2012) that may also explain the strong seismic attenuation observed (De Siena et al., 2017b).

The aforesaid could justify the discrepancy recently highlighted by Moretti et al. (2017) between weak geophysical signals (moderate uplift and low seismicity) and drastic changes in geochemical indicators characterizing the present stage of the CF history.

Finally, it should be mentioned that heterogeneity in the subsoil (Montanaro et al., 2016) and dynamic alterations in subsoil rock matrix properties such as due to seismic energy (Gresse et al., 2016) may modulate emission of stored gas and therefore cause changes in degassing strength.

**5 Conclusions and Perspectives**

About 14 months after the last survey we have revisited the Pisciarelli area. The current $CO_2$ flux was quantified using the portable remote sensing spectrometer LARSS, which detects $CO_2$ in a spatially integrated manner. Although associated with a fairly conservative uncertainty, the result along with fluxes measured in 2016, imply an increase in $CO_2$ flux in the last 2 years. Drawing solid conclusions based on our data is not possible. Nonetheless, given the slow, almost halted ground uplift since 2016, our result could indicate a release of deep magmatic gases towards the hydrothermal system, possibly accompanied by an increased bulk permeability of the shallow crust.

Our measurements, although reasonable, do not permit an unequivocal conclusion whether the origin of the gas emitted at surface is purely hydrothermal or magmatic nor regarding the migration mechanisms from the bottom to the top of the CF plumbing system. Nevertheless, the spatially comprehensive values of $CO_2$ flux acquired through LARSS may help constraining the degassing process as a whole and then provide clues about the strength of the $CO_2$ source, for example via mass balance considerations (Allard et al., 1991) possibly adding to geochemical appraisals (Moretti et al., 2013). However, more measurements of this kind are needed (higher temporal resolution). Furthermore, point measurements should be added in the future to systematically test and verify the capability of LARRS to probe comprehensively all degassing elements in its path. For challenging degassing situations as at CF, integrating LARRS with point measurements may provide a powerful means to obtain a complete picture of degassing. Point measurements are able to draw detailed maps of the emission areas which LARRS is not capable of. However, using two instruments 2D tomography can be performed (Queisser et al., 2016b). Although much more improvement of this technique is needed to converge to degassing maps from point measurements. Moreover, there is potential to further reduce uncertainty of the measured fluxes. To that end, the plume speed estimation will be further improved, especially with respect to resolving the plume speed variations (velocity field) across the scanned plume.

**Author contributions.** M. Queiβer developed LARSS, conducted the measurements and drafted the manuscript, D. Granieri conducted the measurement and drafted the manuscript, M. Burton developed LARSS and drafted the manuscript, F. Arzilli drafted the manuscript, R. Avino and A. Carandente conducted the measurements.

**Data availability.** The data acquired is stored in the University of Manchester's research data repository and may be requested by contacting the corresponding author or mike.burton@manchester.ac.uk.

**Acknowledgements.** The research leading to these results has received funding from the European Research Council Proof-of-Concept Grant CARBSENS (ERC-2016-PoC agreement n. 727626) and starting grant CO2Volc (ERC-SG - ERC Starting Grant agreement n. 279802). We thank G. Tamburello for help with finding a good measurement spot and the staff of the Tennis Hotel Pisciarelli for their cooperation.

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

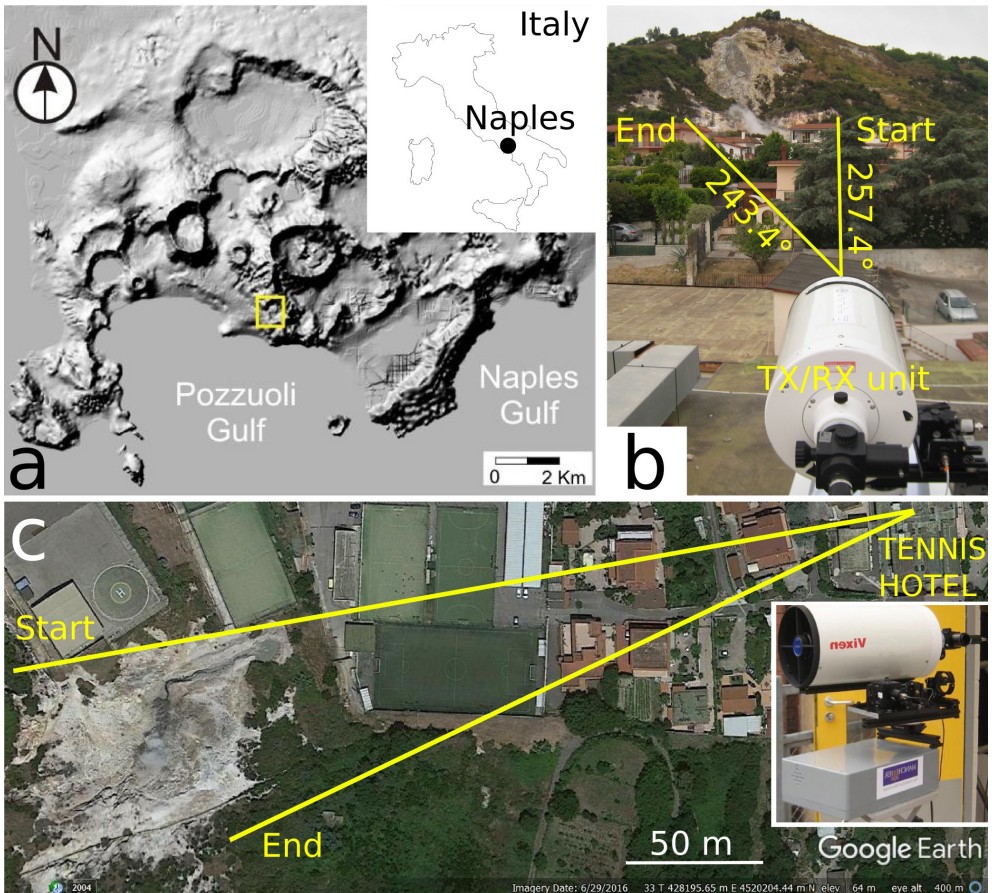

**Figure 1: The location of Campi Flegrei (CF) and the measurement geometry. (a) Map of Italy and relief of the region of CF. The yellow square depicts the zone of Solfatara-Pisciarelli. (b) View from the roof of the Tennis Hotel and the telescope looking towards the Pisciarelli fumarole area concentrated within a zone of ~ 60 m diameter visible in the background. Indicated are the start and the end position of the TX/RX unit's line of sight and the corresponding angles. A total of ~14° was covered during each scan. (c) Nadir view of the situation depicted in (b). The inset shows a complete view of LARSS.**

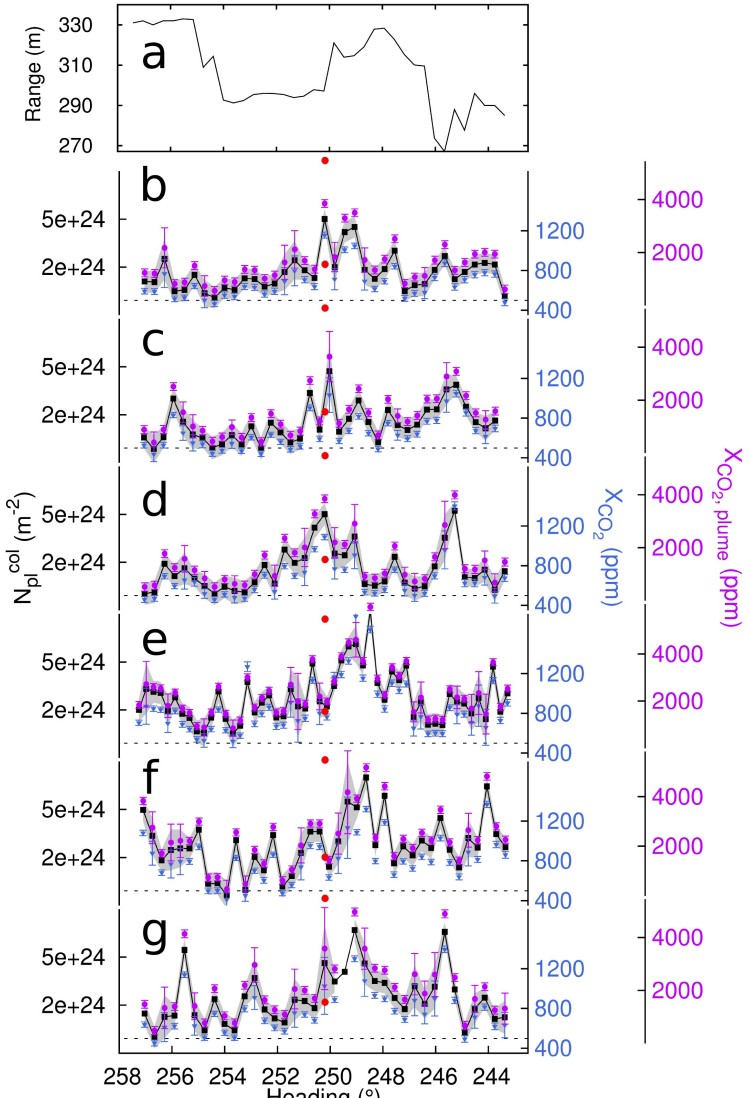

**Figure 2. CO₂ concentration profiles of horizontal scans of the Pisciarelli fumaroles. (a) Ranges to hard target per heading angle. The target was the slope behind the fumaroles (Figs. 1b and c). (b) to (g) Background corrected CO₂ column densities versus angle as used for flux computation [Eq. (1)]. The grey envelope depicts the confidence (1 STD, for details see Queißer et al., 2017). On the right are the corresponding path averaged mixing ratios (blue) with confidence interval (1 STD). The dotted line depicts the ambient CO₂ mixing ratio of 499 ppm. Also shown are the in-plume mixing ratios (magenta), estimated from the path averaged mixing ratios, assuming 62 m plume extension (Queißer et al., 2016a). The red circles mark the minimum and maximum mixing ratio of the same day the measurement took place, registered between 0:00 until midnight in 2h intervals by an in-situ station operated by INGV Naples, located near the center of the scanned area. The scans shown were performed in the order they appear. Their respective acquisition start times were (b to g): 17:15:17, 17:19:43, 17:22:48, 17:39:00, 17:43:47 and 17:52:05.**

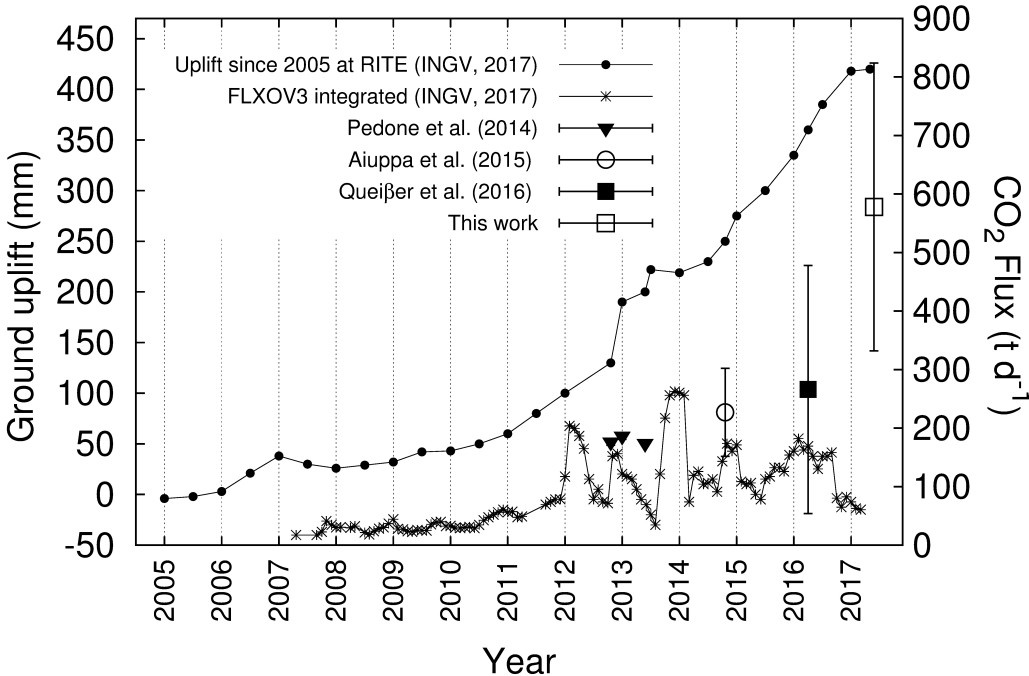

**Figure 3. Ground elevation GPS data from RITE GPS station near the center of CF and $CO_2$ fluxes measured at the Pisciarelli fumarole field. All flux values except FLXOV3 data are spatially integrated. FLXOV3 data is being acquired by an automatic in-situ station in units of g $m^{-2}$ $d^{-1}$. To be comparable to the area-integrated flux values, the data were multiplied with the surface area of the Pisciarelli fumarole area. Two methods of calculating the area yielded very similar results. Approximating the vegetation free area with a polygon yielded 4200 $m^2$, while approximating the surface with a rectangle of dimensions 70 m by 62 m yielded 4340 $m^2$, which was used as it provides a lower limit estimate of the flux.**

| Time of scan | Plume speed $U$ (ms$^{-1}$) | Flux $\Phi_{CO_2}$ (kg s$^{-1}$) |
|---|---|---|
| 17:07:29 | 0.65 ± 0.20 | 6.75 ± 2.63 |
| 17:12:29 | 0.65 ± 0.20 | 5.34 ± 2.72 |
| 17:15:17 (b) | 0.65 ± 0.20 | 4.06 ± 1.98 |
| 17:19:43 (c) | 0.65 ± 0.20 | 3.67 ± 2.18 |
| 17:22:48 (d) | 0.65 ± 0.20 | 3.89 ± 2.06 |
| 17:39:00 (e) | 0.65 ± 0.20 | 8.88 ± 3.24 |
| 17:43:47 (f) | 0.65 ± 0.20 | 8.82 ± 3.29 |
| 17:52:05 (g) | 0.80 ± 0.28 | 8.45 ± 3.56 |
| 17:58:36 | 0.80 ± 0.28 | 10.33 ± 4.00 |
| Mean flux | **6.7 ± 2.9 (578 ± 246 t d$^{-1}$)** | |

**Table 1. Start of the scans performed (local time), the vertical plume speed components with uncertainties (student *t*-deviation) and the corresponding CO$_2$ fluxes with uncertainties (1 STD). Those profiles shown in Fig. 2 have their subfigure identifier written after the time.**