# Peer review of "Increasing CO2 flux at Pisciarelli, Campi Flegrei, Italy"

_Solid Earth, 2017_

## Referee Comment (RC1) · G. Chiodini (Referee) · 24 Jul 2017

The ms is very interesting and merits to be published in Solid Earth discussions. There is just one major point and few minor ones that should be answered by the authors. The main point regards the statement done in the ms that the proposed method is in some way alternative of the classical methods based on the accumulation chambers to monitor the CO2 emissions because, according to the authors, the measurement "... accounts for all possible CO2 vents and diffuse degassing ..... to obtain a quantitative picture of CO2 degassing" In my opinion the proposed method based on "....laser remote sensing spectrometer...." (LARSS) is a very useful additional method to have an almost complete picture of CO2 degassing from an hydrothermal site but the method, at least at this stage of development, can not substitute the accumulation chamber

measurement. LARSS can in fact detect and measure the $CO_2$ emitted by vents, I am not sure that it can reliably measure a real diffuse emission. Diffuse degassing over large areas, such as at Solfatara and Pisciarelli, give rise in fact to some more complex structure than a single plume. So, low level anomalies, that can contribute significantly to the total $CO_2$ release are probably not detectable and quantifiable by LARSS. In addition the method measure the $CO_2$ concentration close to the ground (because the background can not be the sky but the ground) where, for example, the wind field is strongly affected by the interaction of the air with the terrain that implies a reduction in the wind speed etc. This aspect should be a little discussed. Furthermore another aspect of the accumulation chamber method is the possibility to draw detailed maps of the emission areas (and their variation during time), that can not be done with LARSS.

Minor points

- Page 1 line 25 and 28 Substitute d'Auria with D'Auria.

- Page 2 line 4. "...feeding the overlying $\sim$ 1.5 km deep hydrothermal reservoir.." There is any convincing prove of the depth of the hydrothermal reservoir, I suggest to write more generally "...feeding the overlying hydrothermal system(s)...."

- Page 2 line 19 "......Caliro et al., 2014....." Caliro et al., 2014 did not chose any specific depth for magma degassing but they presented a series of different scenarios including degassing fro the 8 km deep (200 Mpa) magma.

- Page 2 line 24 I suggest to substitute alternative with additional.

- Page 2 line 27-29 The cited works refer mainly to the emission of the vent of Pisciarelli. The diffuse degassing eventually included in these measurements is at least incomplete (see main point). I suggest to focus your considerations on the vent emission (that now at Pisciarelli is by far the main way of emission)

- Page 3 line 27-28 " ... The plume speed is retrieved by digital video tracking of the plume of condensed water vapor as described in Queißer et al. (2016).." Ok, the speed

of the plume is measured and it is assumed constant in the plume. Is this assumption reasonable? In my opinion, the colder peripheral zones of the plume should move at a speed lower than the central hot zone. Furthermore a further reduction of the wind speed should be expected in the zones where the plume is just above the terrain (at low height from the ground). In other words I think that this of the speed is still a central parameter with many uncertainties... could you add some discussion about the problem of assuming a constant wind speed?

- Page 4 line 8 Please define what is Delta/beta

- Discussion and Conclusion I agree mostly with you, but I don't think that the Pisciarelli measurements alone could be very indicative without years of monitoring chemical and isotopic compositions of the fumaroles, seismicity and ground deformation. I suggest you to read (and in the case to cite) the most recent paper on Campi Flegrei unrest where the different signals from geochemical and geophysical technique are compared and discussed also in the frame of a physical model of the system (Chiodini et al., 2017). The paper shows further evidence on the pivotal role of the heating of the hydrothermal system in the present dynamic of the caldera. (Chiodini, G., Selva, J., Del Pezzo, E., Marsan, D., De Siena, L., D'Auria, L., Bianco, F., Caliro, S., De Martino, P., Ricciolino, P., and Petrillo, Z., 2017, Clues on the origin of post-2000 earthquakes at Campi Flegrei caldera (Italy): Scientific Reports, doi:10.1038/s41598-017-04845-9)

- References: check the citation of Cardellini et al., 2016, there is an error in the name of one of the coauthors (Giovanni, G. instead of Chiodini, G.)

---

## Referee Comment (RC2) · C. Montanaro (Referee) · 13 Aug 2017

The manuscript report on the use of a very interesting and valuable tool to monitor degassing at the active vents in volcanic area. A portable remote sensing spectrometer LARSS, which detects CO2 in a spatially integrated manner, was used to conduct CO2 flux surveys in Pisicairelli area, located within the Campi Flegrei caldera, Italy. Although measurements are associated with quite few uncertainties, the results indicate an increase in CO2 flux in the last 2 years – findings are well in agreement with other recent study in the area. Based on recent data indicating a deceleration of ground uplift at Campi Flegrei, the authors also suggest that the ongoing degassing it is related to a release of deep magmatic gases towards the hydrothermal system, possibly accompanied by an increased bulk permeability of the shallow crust. Finally, the authors

highlight the importance of the technique in giving spatially comprehensive values of CO2 flux acquired which may help to estimate the degassing process as a whole and then provide clues about the strength of the CO2 source. The paper is very interesting and worthy of publication in Solid Earth discussions, and their results are very important for the understanding of degassing at Pisciarelli, which together with the nearby Solfatara crater, are attracting more and more the scientific attention nowadays. However, the authors attempt a simplified explanation of the degassing behavior while they should consider the complex geology-fluids interaction in the shallow ground (tens of meter) and in the subsoil below the investigated area, which are controlling the surficial degassing. Though the proposed methodology is very valuable, it should also be considered (and discussed in the manuscript) that its integration with other punctual measurements techniques (e.g. accumulation chamber) is needed to better characterize the areal degassing and constrain the effect of local elements (e.g. fractures) on the degassing behavior.

Thus, I would suggest minor revisions on the following points in the manuscript:

- Page2, lines 3-6: here the recent work on the geology and the structure of the area should be mentioned (Isaia et al. 2015 and Vitale et al. 2014):

- Page2, lines 9-10: here I would also discuss the effect of i) the subsoil in controlling the surficial degassing (Montanaro et al. 2016), and ii) passing of a seismic wave that can induce a strong increase in the total amount of gas (Gresse et al. 2016);

- Page2, lines 20-21 (and in the discussion as well): the recent work of Mayer et al. (2016) and Piochi et al. (2015), concerning the effect solfataric alteration that increases porosity and permeability of altered rock, should be mentioned and discussed;

- In "Materials and methods": maybe here should be briefly discussed about other factors influencing the measurements, such as wind, change in humidity around the measured spots, etc., which are also mentioned in the results;

- Page4, line 10: "gas plume" rather than volcanic;

- Page5, line 14: "td-1" rather than "kgs-1"(?);

- Page6, line 8-9: here the works of Vanorio (2015) and Heap (2014) on the properties of the caldera-filling tuffs should also be cited and maybe briefly discuss about it.

- Figure 2: can you reverse the Heading angle values in a way that is consistent with Figure 1B?

Here the suggested citations:

Vitale S, Isaia R. Fractures and faults in volcanic rocks (Campi Flegrei, southern Italy): Insight into volcano-tectonic processes. International Journal of Earth Sciences. 2014 Apr 1;103(3):801-19.

Isaia R, Vitale S, Di Giuseppe MG, Iannuzzi E, Tramparulo FD, Troiano A. Stratigraphy, structure, and volcano-tectonic evolution of Solfatara maar-diatreme (Campi Flegrei, Italy). Geological Society of America Bulletin. 2015 Sep 1;127(9-10):1485-504.

Gresse M, Vandemeulebrouck J, Byrdina S, et al (2016) Changes in CO2 diffuse degassing induced by the passing of seismic waves. J Volcanol Geotherm Res 320:12–18. doi:10.1016/j.jvolgeores.2016.04.019

Mayer K, Scheu B, Montanaro C, et al (2016) Hydrothermal alteration of surficial rocks at Solfatara (Campi Flegrei): Petrophysical properties and implications for phreatic eruption processes. J Volcanol Geotherm Res 320:128–143. doi: 10.1016/j.jvolgeores.2016.04.020

Piochi M, Mormone A, Balassone G, Strauss H, Troise C, De Natale G (2015) Native sulfur, sulfates and sulfides from the active Campi Flegrei volcano (southern Italy): Genetic environments and degassing dynamics revealed by mineralogy and isotope geochemistry, Journal of Volcanology and Geothermal Research, Volume 304, 2015, Pages 180-193, ISSN 0377-0273, http://dx.doi.org/10.1016/j.jvolgeores.2015.08.017.

Heap MJ, Baud P, Meredith PG, et al (2014) The permeability and elastic moduli of tuff from Campi Flegrei, Italy: Implications for ground deformation modelling. Solid Earth 5:25–44. doi: 10.5194/se-5-25-2014

Vanorio T, Kanitpanyacharoen W, (2015) Rock physics of fibrous rocks akin to Roman concrete explains uplifts at Campi Flegrei Caldera, Science, DOI: 10.1126/science.aab1292

Montanaro C, Mayer K, Scheu B, Isaia R, Mangiacapra A, Gresse M, Vandemeulebrouck J, Moretti R, Dingwell DB. Hydrothermal activity and subsurface soil complexity: implication for outgassing processes at Solfatara crater, Campi Flegrei caldera. InEGU General Assembly Conference Abstracts 2016 Apr (Vol. 18, p. 12509).

---

## Editor Comment (EC1) · M. J. Heap (Editor) · 13 Aug 2017

Dear Manuel Queißer,

As you have no doubt seen, I have now received two reviews of your manuscript "Increasing CO2 flux at Pisciarelli, Campi Flegrei, Italy". Both reviewers offer constructive comments and recommend publication following revision. Please now prepare a revised manuscript and point-by-point rebuttal letter. Please pay particular attention to the comments by reviewer #2 related to the "pros and cons" of the LARSS method. I look forward to reading your revised manuscript.

Thanks,

Mike Heap (Topical Editor of Solid Earth)

---

## Author Comment (AC1) · 29 Aug 2017

Authors response to review of EGU SE manuscript 2017-70

The authors would like to express their gratitude to the time invested by the two reviewers and the editor. Below you find a point to point response to the issues raised by the reviewers. Both reviewers pointed out that LARRS cannot replace in situ measurement but is a complementary technique. This has been amended accordingly in the manuscript. Changes are highlighted in yellow. As they are interrelated, we decided to put the responses to both reviewer comments in this one document.

Reviewer #1

**General comments**

**The ms is very interesting and merits to be published in Solid Earth discussions. There is just one major point and few minor ones that should be answered by the authors. The main point regards the statement done in the ms that the proposed method is in some way alternative of the classical methods based on the accumulation chambers to monitor the CO2 emissions because, according to the authors, the measurement "... accounts for all possible CO2 vents and diffuse degassing ..... to obtain a quantitative picture of CO2 degassing" In my opinion the proposed method based on "....laser remote sensing spectrometer...." (LARSS) is a very useful additional method to have an almost complete picture of CO2 degassing from an hydrothermal site but the method, at least at this stage of development, can not substitute the accumulation chamber LARSS can in fact detect and measure the CO2 emitted by vents,**
Reply:
We agree. The precision of LARRS at the moment does not allow this. In situ measurements are much more precise and accurate. However, as it measure path integrated, everything within the path contributes, meaning that LARRS may help to give representative measurements from large regions, the more its precision improves the more it will do so.
Changes: In abstract: Thanks to the integrated path soundings, LARSS may help to give representative measurements from large regions containing different $CO_2$ sources (…).

Added p3, l2:
On the other hand, point measurements are very precise and valuable in characterizing local degassing elements, such as

fractures. Path integrating, scanning gas measurement techniques, on the other hand, may add value by providing a spatially comprehensive measurement. To attempt a spatially inclusive measurement of all possible sources of $CO_2$, diffusive soil and vented degassing, we used a laser remote sensing spectrometer (LARSS), developed in the ERC proof-of-concept project CarbSens. Combined with point measurement techniques, such as accumulation chambers, LARRS may help to yield a more complete picture of degassing.

Added in conclusions:
Furthermore, point measurements should be added in the future to systematically test and verify the capability of LARRS to probe comprehensively all degassing elements in its path. For challenging degassing situations as at CF, integrating LARRS with point measurements may provide a powerful means to obtain a complete picture of degassing.

**I am not sure that it can reliably measure a real diffuse emission. Diffuse degassing over large areas, such as at Solfatara and Pisciarelli, give rise in fact to some more complex structure than a single plume. So, low level anomalies, that can contribute significantly to the total CO2 release are probably not detectable and quantifiable by LARSS.**
Reply: As said above, if the low level anomaly is in the measurement path and the precision is high enough (high SNR), it will be detected, why shouldn't it?

**In addition the method measure the CO2 concentration close to the ground (because the background can not be the sky but the ground) where, for example, the wind field is strongly affected by the interaction of the air with the terrain that implies a reduction
in the wind speed etc. This aspect should be a little discussed.**
Reply: Added at p5 l10: Given the complex terrain and the fact that the measurement was performed close to the ground the velocity field across the scanned plume was generally not constant, in addition to temperature variations causing different plume speeds across the plume. The corresponding variability has been accounted for by tracking different paths of propagating water vapor across the plume and using the variability in the error estimation. Plume speed is in fact one of the main sources of uncertainty, adding an uncertainty of the order of 30% to the flux.
In conclusions: To that end, the plume speed estimation will be further improved, especially with respect to resolving the plume speed variations (velocity field) across the scanned plume.

**Furthermore another aspect of the accumulation chamber method is the possibility to draw detailed maps of the emission areas (and their variation during time), that can not be done with LARSS.**

Reply: Not with one instrument, but with two you can do tomography. It is still unmature but it will improve. We have added this in the conclusions: Point measurements are able to draw detailed maps of the emission areas which LARRS is not capable of. However, using two instruments 2D tomography can be performed (Queisser et al., 2016b). Although much more improvement of this technique is needed to converge to degassing maps from point measurements.

**Specific comments**

**- Page 1 line 25 and 28 Substitute d'Auria with D'Auria.**
Done

**- Page 2 line 4. "...feeding the overlying ~ 1.5 km deep hydrothermal reservoir.." There is any convincing prove of the depth of the hydrothermal reservoir, I suggest to write more generally "...feeding the overlying hydrothermal system(s)...."**
Changed accordingly

**- Page 2 line 19 "......Caliro et al., 2014....." Caliro et al., 2014 did not chose any specific depth for magma degassing but they presented a series of different scenarios including degassing fro the 8 km deep (200 Mpa) magma.**
This section is not meant to split the world in two sides but it is just for the unoccupied reader to find some more information on the subject. Caliro et al. Is just an informative reference. We prefer to leave it.

**- Page 2 line 24 I suggest to substitute alternative with additional.**
changed.

**- Page 2 line 27-29 The cited works refer mainly to the emission of the vent of Pisciarelli. The diffuse degassing eventually included in these measurements is at least incomplete (see main point). I suggest to focus your considerations on the vent emis-**
**sion (that now at Pisciarelli is by far the main way of emission)**
Confronting point with spatial measurements is one of the main points of this paper, including Fig. 3. While we agree that LARRS

cannot replace point measurements and amended the MS accordingly (see above), LARRS is another way of probing degassing adding some value and we prefer to cite these works as they use a similar technique than LARRS and indeed measured at Pisciarelli (see Fig. 3). Accordingly, we changed

A spatially comprehensive  measurement of $CO_2$ flux that accounts for all possible $CO_2$ vents and diffuse degassing is desirable to obtain a quantitative picture of $CO_2$ degassing, but has only been done a few times after 2012 at Pisciarelli (Pedone et al., 2014; Aiuppa et al., 2015; Queißer et al., 2016a).

to

A spatially integrated measurement of $CO_2$ flux has only been done a few times after 2012 at Pisciarelli (Pedone et al., 2014; Aiuppa et al., 2015; Queißer et al., 2016a).

**- Page 3 line 27-28 " ... The plume speed is retrieved by digital video tracking of the**
**plume of condensed water vapor as described in Queißer et al. (2016).." Ok, the speed of the plume is measured and it is assumed constant in the plume. Is this assumption reasonable? In my opinion, the colder peripheral zones of the plume should move**
**at a speed lower than the central hot zone. Furthermore a further reduction of the**
**wind speed should be expected in the zones where the plume is just above the terrain**
**(at low height from the ground). In other words I think that this of the speed is still a**
**central parameter with many uncertainties... could you add some discussion about the**
**problem of assuming a constant wind speed?**
Reply: Please see reply to one of your general comments above: In addition the method measure the CO2 concentration close to the ground …

**- Page 4 line 8 Please define what is Delta/beta**
It is defined on page 3 l 25.

**- Discussion and Conclusion I agree mostly with you, but I don't think that the Pisciarelli measurements alone could be very indicative without years of monitoring chemical and isotopic compositions of the fumaroles, seismicity and ground deformation. I suggest you to read (and in the case to cite) the most recent paper on Campi Flegrei unrest**

**where the different signals from geochemical and geophysical technique are compared and discussed also in the frame of a physical model of the system (Chiodini et al., 2017). The paper shows further evidence on the pivotal role of the heating of the hydrothermal system in the present dynamic of the caldera. (Chiodini, G., Selva, J., Del Pezzo, E., Marsan, D., De Siena, L., D'Auria, L., Bianco, F., Caliro, S., De Martino, P., Ricciolino, P., and Petrillo, Z., 2017, Clues on the origin of post-2000 earthquakes at Campi Flegrei caldera (Italy): Scientific Reports, doi:10.1038/s41598-017-04845-9)**

Reply: In discussion we added: Recent findings indeed point towards an impulsive influx of hot magmatic fluids into the hydrothermal system as a possible source mechanism at CF that eventually cause the observed geophysical and geochemical time series, including the present one (Chiodini et al., 2017).

**- References: check the citation of Cardellini et al., 2016, there is an error in the name of one of the coauthors (Giovanni, G. instead of Chiodini, G.)**

Reply. Corrected

Referee #2

**General comments**

**The manuscript report on the use of a very interesting and valuable tool to monitor degassing at the active vents in volcanic area. A portable remote sensing spectrometer LARSS, which detects CO2 in a spatially integrated manner, was used to conduct CO2 flux surveys in Pisicairelli area, located within the Campi Flegrei caldera, Italy. Although measurements are associated with quite few uncertainties, the results indicate an increase in CO2 flux in the last 2 years – findings are well in agreement with other recent study in the area. Based on recent data indicating a deceleration of ground uplift at Campi Flegrei, the authors also suggest that the ongoing degassing it is related to a release of deep magmatic gases towards the hydrothermal system, possibly accompanied by an increased bulk permeability of the shallow crust. Finally, the authors highlight the importance of the technique in giving spatially comprehensive values of CO2 flux acquired which may help to estimate the degassing process as a whole and then provide clues about the strength of the CO2 source. The paper is very interesting and worthy of publication in Solid Earth discussions, and their results are very important for the understanding of degassing at Pisciarelli, which together with the nearby Solfatara crater, are attracting more and more the scientific attention nowadays.**

**However, the authors attempt a simplified explanation of the degassing behavior while they should consider the complex geology-fluids interaction in the shallow ground (tens of meter) and in the subsoil below the investigated area, which are controlling the surficial degassing.**
Reply: Given the very limited data we obtained, any deeper insights in the source mechanism is utterly out of scope of this paper. The discussion is meant to relate our findings to other geomechanical, geophysical and geochemical observations. We cannot provide an explanation based on this paper and we say so in the introduction and the conclusions.

**Though the proposed methodology is very valuable, it should also be considered (and discussed in the manuscript) that its integration with other punctual measurements techniques (e.g. accumulation chamber) is needed to better characterize the areal degassing and constrain the effect of local elements (e.g. fractures) on the degassing behavior.**
Exactly this point has been raised already by reviewer 1 and treated accordingly. Please see our reply to his comments above.

**Specific comments**

Thus, I would suggest minor revisions on the following points in the manuscript:

**- Page2, lines 3-6: here the recent work on the geology and the structure of the area should be mentioned (Isaia et al. 2015 and Vitale et al. 2014):**

We added Vitale and Isaia, 2014.

**- Page2, lines 9-10: here I would also discuss the effect of i) the subsoil in controlling the surficial degassing (Montanaro et al. 2016), and ii) passing of a seismic wave that can induce a strong increase in the total amount of gas (Gresse et al. 2016);**

Reply: While we agree that these effects are important in modulating the degassing strength, the focus of this paper is more on the actual source of the $CO_2$ and the mechanisms which control it at the first place. But we deem it important to mention it in the discussion. Added both references: Finally, it should be mentioned that heterogeneity in the subsoil (Montanaro et al., 2016) and dynamic changes in subsoil rock matrix properties (Gresse et al., 2016) may modulate emission of stored gas.

**- Page2, lines 20-21 (and in the discussion as well): the recent work of Mayer et al. (2016) and Piochi et al. (2015), concerning the effect solfataric alteration that increases porosity and permeability of altered rock, should be mentioned and discussed;**

Reply: We already mention an increase in permeability. Adding this petrological results just leads to far away from the scope of this paper, which is not to explain why the permeability increased. For a review paper yes, but here we prefer to not discuss this. But we added a reference to Piochi on p2 l 20.

**- In "Materials and methods": maybe here should be briefly discussed about other factors influencing the measurements, such as wind, change in humidity around the measured spots, etc., which are also mentioned in the results;**

Wind, i.e. plume speed is treated in the response to reviewer #1. Humidity plays a role when you convert $CO_2$ number density to mixing ratio (which is relative to the total number of air molecules hence including water), but this is not done here to compute fluxes. Mixing ratios are only shown for display purposes. That said, the difference between dry and wet air mixing ratio is negligible

compared with the uncertainty we get from the plume speed estimation, for example.

**- Page4, line 10: "gas plume" rather than volcanic;**
changed to gas plume

**- Page5, line 14: "td-1" rather than "kgs-1"(?)**;
No it is indeed kg s-1.

**- Page6, line 8-9: here the works of Vanorio (2015) and Heap (2014) on the properties of the caldera-filling tuffs should also be cited and maybe briefly discuss about it.**
We measure an increase in $CO_2$ output and think this result is robust because the method we use gives comprehensive $CO_2$ concentrations. That is all we say. Second order complication is relation this to other data and modeling results. Adding above work would cause third order detail branching that is out of scope of this short comm paper.

**- Figure 2: can you reverse the Heading angle values in a way that is consistent with Figure 1B?**
The headings are absolute values relative to north, that is why they are displayed in descending order. We prefer to leave it that way as this is was the direction of the scan.